

# Tsunami deposits in Martinique related to the 1755 Lisbon earthquake

Valérie Clouard[1], Jean Roger[2], and Emmanuel Moizan[3]

[1]Observatoire Volcanologique et Sismologique de Martinique, Institut de Physique du Globe de Paris (IPGP), Martinique, FWI.

[2]Université des Antilles, Laboratoire LARGE, Campus de Fouillole, 97159 Pointe-à-Pitre, Guadeloupe, now at G-MER Etudes Marines, Avenue de l'Europe, 97118 Saint-Francois, Guadeloupe, FWI.

[3]Institut National de Recherches Archéologiques Préventives, Centre de recherche archéologique de Poitiers, 122, rue de la Bugellerie, 86000 Poitiers, France

*Correspondence to:* Clouard (clouard@ipgp.fr)

**Abstract.** In order to assess tsunami hazard in oceanic islands, one needs to enlarge the observational time window by finding more evidence of past events. To that end, evidence of allochthonous deposits provides estimates of tsunami inundation, recurrence time and magnitude. However, in tropical islands, erosion due to the highly rainy climate generally prevents deposits to stay in place and when they are, relating them to a tsunami is not straightforward, as they can result either from a strong
hurricane or from a tsunami. One notable exception concerns deposits sealed by subsequent events. In this paper, we present evidence of an anomalously thick two-layer tsunami deposit in an excavation in Martinique. Analysis of the archaeological remains indicate that it is related to the 1755 Lisbon tsunami. We explain the thickness of the deposit by a tsunami-induced bore in the mangrove drainage channels of Fort-de-France. Our results highlight the benefits of collaborative research involving geology and archaeology, indicate a way to improve our tsunami databases and further constrain the use of numerical modelling
to predict paleo-tsunami deposit thickness.

## 1 Introduction

The Lesser Antilles volcanic arc has been formed since Eocene by the subduction of the North American plate beneath the Caribbean plate (Bouysse and Westercamp, 1990), with a low convergence rate (ca. 2 cm/yr, DeMets et al., 2000). However, this subduction has the potential to generate moderate to large thrust type earthquakes (Feuillet et al., 2011; Hough, 2013),
potentially tsunamigenic (Hayes et al., 2013). During historical times (about 500 yrs), 2 large thrust earthquakes occurred in the Lesser Antilles, one located east of Martinique (1839) and the other one east of Guadeloupe (1843) (Dorel, 1981; Bernard and Lambert, 1988). Although recent studies estimate the 1843 earthquake magnitude to be > 8.5 (Feuillet et al., 2011; Hough, 2013; Hayes et al., 2013), a magnitude with the potential to generate a tsunami, no significant tsunamis generated by the Lesser Antilles subduction have been documented during historical times. For the 1843 event, historical records only contains the
description in Antigua of a brief 4-foot rise followed by an immediate descent in sea level (Bernard and Lambert, 1988). For the 1839 earthquake, we find the first description of an anomalous behavior of the sea in "Le chapelier pirate"(Peray, 1843,





p.327) saying "Plus terrible encore, la mer bouleversée se présenta dans une position déversante sur toute la ville et eut le temps de gagner dix à quinze mètres sur elle, mais, fort heureusement, le mouvement se fit en sens inverse plus promptement que par la parole...", which can be translated as "Even more terrible, the shook up sea rose to cover all the city in a pouring position and had time to gain ten to fifteen meters on it, but, very fortunately, the sea withdraw more quickly than time allowed...".

More generally in the Lesser Antilles, tsunami catalogues (Zahibo and Pelinovsky, 2001; Lander et al., 2002; Dunbar and McCullough, 2012) indicate about ten historical tsunamis related to regional or trans-oceanic earthquakes (1690, 1755, 1767, 1802, 1823, 1824, 1831, 1843, 1969, 1985, 2004) , sometimes with run-up and wave information, but all with small to moderate run-up and very few damages.

During historical times, no noticeable tsunami was observed in Martinique except the 1755 one (Roger et al., 2011). So far, the 1st of November 1755 M 8.5-9.0 Lisbon earthquake triggered the most important known transoceanic tsunami that travelled across the Northern Atlantic Ocean (e.g., Baptista et al., 2003). According to many coeval historical reports, cross-fitted with numerical modeling results, this tsunami is well-known to have reached Portugal, Spain, Morocco, Great Britain, Azores, Madeira, Newfoundland, Bermuda, the Lesser Antilles, Cuba and Brazil (Fig. 1 and Table 1 in Electronic Supplement), with observed waves up to several meters. However, very few deposits can be attributed with certainty to this particular event, even at near-field locations, and none in the western Atlantic countries (Fig. 1). It is related to dating difficulties in particular for recent tsunami deposits: dating uncertainties on sediments using radiocarbon or luminescence dating or a combination of several methods can reach 20% to 125% (e.g., Cunha et al., 2010; Cuven et al., 2013), and similar large uncertainties are obtained when dating erosional structures (Rebêlo et al., 2013).

Based on tectonic and tsunami catalogues, Parsons and Geist (2008) found that the highest tsunami probabilities in the Caribbean for a tsunami run-up to be >0.5m are for the Lesser Antilles Islands. However, no historical nor paleo-tsunami deposits have ever been identified on-land in our studied area. Generally, a significant tsunami (which we arbitrarily define as more than 1 m high) reaching shallow waters is able to produce extensive erosion and to carry a lot of sediment of various grain sizes, and including boulders. This material can be ripped away from the sea bottom and from the coast by the tsunami and deposited onshore during the flow (Scheffers and Kelletat, 2003; Paris et al., 2007; Goto et al., 2010). With time, deposits are buried and the stratigraphic analysis of trenches or cores exhibits a characteristic pattern of alternation of soil and tsunami deposits as seen in Chile (Cisternas et al., 2005) or Japan (Sawai et al., 2009), whereas boulders can eventually outcrop. Buried sandy layers and displaced reef rock boulders are thus commonly used to identify overwash events. But even when a complete sequence of deposits is found, particularly in tropical islands, they can result from a tsunami or from a storm surge (Morton et al., 2007; Spiske et al., 2008), as mentioned for Anegada boulders (Atwater et al., 2012; Buckley et al., 2012), in British Virgin Islands. Some large boulders, dated 2500-2700 years before present, have been found on the east coast of Guadeloupe and attributed to a single tsunami, nevertheless a stormy origin is difficult to rule out (Scheffers et al., 2005). Finally, it is unlikely that tsunami deposits are preserved as they were formed (Bahlburg and Spiske, 2012), depending on various factors, such as tsunami dynamics, landforms, sediment transport, climatic conditions, erosion processes, coastal environment and of human activity through urban and agricultural pressure.



In this paper, we present a 8 cm-thick, two-color layer of sand found in an archaeological excavation carried out in 2012 in the center of Fort-de-France, Martinique. From historical data, we infer an origin related to 1755 Libon tsunami. We explain its two-color aspect and its unusual thickness by the effect of the propagation of the tsunami waves in the drainage channel of Fort-de-France.

## 2 Sandy layer in an archaeological excavation in Martinique

In November 2012, the Institut National de Recherches Archéologiques Préventives (Inrap) started an archaeological excavation on the location of the future Court of Appeal in the historical center of Fort-de-France. The 2500 m2 area reveals remains of successive buildings (Fig. 2). The very first building, from the XVII century, appears on the land registry of 1726 (Raussain, 1726). It was a L-shape building, the backyard being merely a garden. By the middle of the XVIII century, a new U-shape building open eastward replaced the first edifice (Fig. 2c). In direct contact with the archaeological levels associated with this new construction, an 8 cm-thick sandy layer is present. It consists of a ∼1 cm-thick rich shelly lighter-colored layer at its base and an upper 6-9 cm-thick black layer with no apparent grading (Fig. 3).

This deposit is located primarily in the garden in the south-east area of the site, and in the new southern annexes (Fig. 2a). No sand is found inside the rooms of the southern and central part of the main building, except in front of the door, whereas sand is accumulated at the bottom of the walls elsewhere. The northern wing of the building is completely covered by sandy deposits. The stratigraphic cross section of the excavation (Fig. 2) indicates that in a first stage, the XVII century building walls and ground were surfaced, and then covered by white sand and mortar embankment. At this precise moment, a sandy layer sealed the whole area, including the new foundations, and entered the opened rooms. Field observations indicate that the construction was stopped for a while (days or weeks), and when the construction resumes, the sandy layer is removed only where the new foundations are built and the XVIII century building is constructed. It is finished during the second part of the XVIII century (De Bexon, 1782).

The 8 cm-thick deposit presents clearly defined up and down limits, confined between light colored embankment materials and is composed of two different layers (Fig. 3). The bottom thin light colored layer (1 to 2 cm) of broken shells and white sand is deposited everywhere on the site and follows the irregularities of the erosive contact. The upper black layer is mainly composed of coarse sand of volcanic origin, but include numerous marine shells and rounded pebbles. A terrestrial origin, as a result of a lahar flow or river flood, is excluded based on the fact that the mineralogical type of the deposit exhibits a serious lack of terrestrial carbon materials, showing at most an occasional small piece of wood. The physical aspect of the deposit, and the fact that Fort-de-France's Bay is very well protected against storms and their associated waves and surges, also excludes a stormy origin of these sandy layers. It is confirmed by the lack of large hurricanes in Martinique between 1713 and 1780 (Romer, 1932). The lower limit of the deposit is irregular and erosive, which corresponds to the erosive action of an incident flow before any depositional process begins. The black layer globally follows the light grey layer, indicating a coeval deposition. Its upper limit is a bit diffusive, probably due to the latter manual spreading of the embankment materials observed in the excavation. In general, tsunami deposit comes from inflow and backflow origin. In our case, the geometry of



the site, open eastward and closed westward, and the sand particles sub-horizontal lamination (Fig. 3c) shows that the water enters the site and the sediments deposit slowly, and that there is no contribution from upper backflow. In summary, these two superimposed layers of distinct origin can clearly be associated with a single tsunami event.

## 3 Origin of the Martinique tsunami deposit

The temporal origin of the tsunami ranges between 1726 and 1783. During this period, Martinique was stroke by two tsunamis, in 1755 and 1767. The $1^{st}$ of November 1755 M~8.5-9.0 Lisbon earthquake triggered the most important known transoceanic tsunami that travelled across the Northern Atlantic Ocean (e.g., Baptista et al., 2003). According to many coeval historical reports, cross-fitted with numerical modeling results, this tsunami is well-known to have reached Portugal, Spain, Morocco, Great Britain, Azores, Madeira, Newfoundland, Bermuda, the Lesser Antilles, Cuba and Brazil, with observed waves up to
several meters. In Martinique, numerous historical records report 1 to 3-m height waves in all the coastal areas, including Fort-de-France (Roger et al., 2011). Respectively, the Barbados earthquake of April $24^{th}$, 1767 generated a local tsunami (O'Loughlin and Lander, 2003; Lander et al., 2003) and 3-feet waves were observed only on the eastern coast of Martinique (see a newspaper extract in http://tsunamis.brgm.fr). It is doubtful that the impact of this latter event, which occurred only 12 years after the notable 1755 tsunami, would not have been reported in Fort-de-France. We can thus relate with certainty the
Fort-de-France deposits a unique event, the 1755 Lisbon tsunami.

However, two questions remained unanswered concerning on one hand the origin of the 2-layer sandy materials, and on the other hand, the thickness of the deposit (~8 cm) related to a ~1 m historically reported and modeled tsunami (Roger et al., 2011). The spatial distribution of the tsunami deposit thickness after the Tohoku-oki earthquake shows that a 10-cm deposit correspond to a 2.5-3 m tsunami (Takashimizu et al., 2012). Partially, the lack of backflow, which has an erosive effect on
the deposited sand (Furusato and Tanaka, 2014), contributes to the thickening of our deposit. The thin lightly-colored layer at the basement of the deposit can be attributed to the bottom of Fort-de-France's Bay, which presently exhibits the same kind of materials. It was transported by the tsunami waves straight to the town. The origin of the black material, and its thickness are more puzzling. Comparing this black sand to the different samples we collected afterwards from neighboring beaches and in the upper Madame river, we find a high similarity rate in terms of mineralogical composition and grain sizes (Fig. 4) only
with the sand coming from a beach located just westward to the Madame river mouth (located on Fig. 4). In May 2014, a preliminary archaeological finding was observed at site 2 located  250 m northeastward from the first one (see location on Fig. 4): among the 8 trenches that were dug and immediately re-filled, a sandy layer, similar to the one of site 1 was present in 6 of them (A. Jegouzo, pers. comm). During the XVIII century, Fort-de-France has been constructed on a mangrove and numerous channels were present so they could drain the water from the city (Fig. 4). As it has been observed in many places (Peters et al.,
2007; Tanaka et al., 2012; Ely et al., 2014) and during the recent tsunamis, the upriver propagation of tsunami waves can be amplified by a tsunami induced bore (Chanson and Lubin, 2013). In the case of the 1755 tsunami inundation of Fort-de-France, we propose that the Madame's river mouth ridge was removed by the tsunami-induced bore and that the upstream propagation





subsequently flooded the center of Fort-de-France, using the draining channels indicated on contemporaneous historical maps (Fig. 4). This phenomenon explains both the 2-layer deposit and the abnormal thickness of the dark sand layer.

## 4 Discussion

In the specific context of tropical volcanic islands, coastal areas are constantly affected by erosion from the sea (waves and
wind) and from the land (river overflows, landslides, bioturbation and anthropogenic activities). In tropical climate, most of the year and especially during the cyclonic period, important orographic rainfalls increase the soil erosion and the river flooding. This results in absent or weak tsunami deposits, which most probably will be destroyed afterward by the human activity. In Martinique, we found that most of the present-day coastal cities are built on former mangrove forests, which were completely banked up to meet the growing need for housing the increasing population in this small montinous island. In addition, hurricanes
are rather frequent in comparison with tsunamis, and are also able to inundate low-lying areas and to produce similar deposits (Shanmugam, 2012). For these reasons, not only do these processes make it difficult to find deposits of marine submersion in the Lesser Antilles, but when found, it is extremely difficult to determine their origin, as it has been shown in Anegada, British Virgin Islands (Atwater et al., 2012, 2017).

Nevertheless, we have found in Fort-de-France conclusive evidence of 1755 Lisbon tsunami deposits. The 1755 Lisbon
tsunami has been widely reported in the whole Caribbean region as previously mentioned. In Anegada, Atwater et al. (2017) propose that an important sheet of sand and shells dates to the interval 1650-1800 can be related to the 1755 Lisbon tsunami. In Fort-de-France, a 0.9 m level elevation in the Madame river has been observed and is correctly reproduced by numerical modeling (Roger et al., 2011) : our deposition site is located within the modeled inundation zone of the 1755 tsunami (Fig. 4). But our results bring a new restriction to use of the deposit thickness as a proxy for tsunami intensity estimation. Our
thin lower layer corresponds to the normal up-stream sediment transportation due to the incoming tsunami waves bringing sediment straight from the offshore sea-bottom. The thick upper deposit was produced subsequently by upriver propagation of the tsunami, as observed during the recent tsunamis of Chile (Fritz et al., 2011) and Japan (Mori et al., 2011), and also seen in the numerical modeling of the 1755 tsunami in Lisbon (Baptista et al., 1998). This propagation in a narrow channel is highly turbulent and able to carry a large amount of suspended sediment (see review in Chanson and Lubin, 2013). In addition,
the specific geometry of the Cour d'Appel building prevent the outflow from eroding the inflow deposit as also observed in Kuril islands (MacInnes et al., 2009). It is interesting because the observed deposit thickness is commonly used to calibrate the tsunami propagation model and estimate the intensity of paleo-tsunamis (e.g., Jaffe and Gelfenbaum, 2007; Jaffe et al., 2012). Recently, in the context of near-field earthquakes, Goto et al. (2012) have also shown that the liquefaction produced by the strong ground motion is responsible for 4/5 of the tsunami deposits, whereas only 1/5 comes from beach erosion. In the context
of far-field earthquakes, our study indicates that about 7/8 of the deposits can come from a river bore, and only 1/8 from the beach or bay erosion. Thus, a careful analysis of the tsunami deposit thickness and origin must be a prerequisite before any numerical modeling based on the tsunami deposit thickness. Conversely, the observed thickening of the tsunami deposits for a ∼1m tsunami in case a river crosses the city provides important constraints on hazard assessment in densely populated areas.



It was a fortuitous chance for us to find undisturbed tsunami deposits buried in a town that has been modified so many times during its short history. But it is likely that this can occur in other coastal American cities. The urban development of many presently large cities in the Americas began during the XVIII century, and it is plausible that the 1755 tsunami deposits can also be found in locations in eastern coastal American cities by investigating historical building sites that may have sealed
sediments from this event. Further, as the conservation of tsunami deposits are rare in natural context, we can also hypothesize that collaborative geological and archaeological studies of pre-Colombian sites could also reveal paleo-tsunami deposits and enable us to enlarge our observation window and improve our tsunami catalogues. As demonstrated for archaeoseismology (Nur, 2007), archaeological analysis appears to be a powerful tool to precisely date paleo-tsunamis within a range of dates (∼several months to years) that is not reachable by classic dating methods, especially for the last centuries.

## 5  Conclusions

We provide the first conclusive observation of the 1755 Lisbon tsunami deposits in the Americas. The 6-8 cm observed thickness of the sedimentary layers is abnormally large to have been produced by a ∼1 m tsunami. It can be explained by 2 successive floodings, one related to the direct tsunami front waves and responsible for about 1/8 of the deposit, and the other one produced by a tsunami-induced bore. This indicates that the tsunami deposit thickness used in tsunami modeling is a parameter
that must be carefully checked in order to avoid overestimation of paleo-tsunamis and to correctly assess the tsunami hazard. Tsunami hazard assessment is of primary importance for oceanic islands because of the highly populated coastal area linked to increasing tourism development, the concentration of infrastructure in low-lying areas and the expectation of sea-level rise associated with climate change. Our results indicate that the use of all available geological methodologies and collaborative studies with historians and archeologists might enable us to improve our historical tsunami catalogs in the future, thus helping
the preparedness of tsunami hazard plans for coastal communities.

*Acknowledgements.* This work has been partially funded by the INTERREG IV Tsunahoule project and by the INTERREG IV Tsuareg project. Part of Fig. 3 has been designed by D. Billon, according to topographic leveling made by A. Daussy. The authors are grateful to S. Tait and C. Moore for their kind help to improve English language. We thanks J.N. Degrace who sent us the Martinique hurricane catalogue of Romer and A. Jegouzo for her valuable comments on the site 2 excavation. This is an IPGP contribution #3871.



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





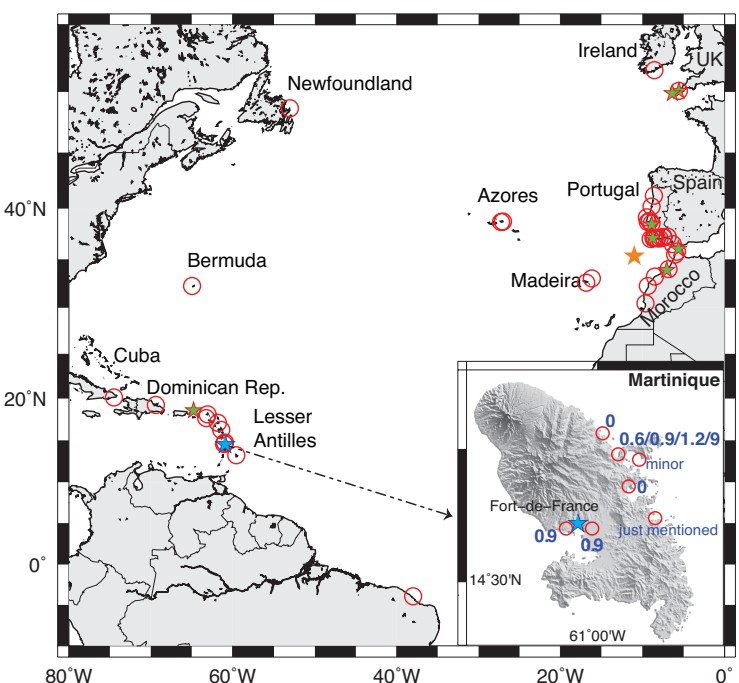

**Figure 1.** Location of the 1755 tsunami historical records (red circles) and of the deposits that could be related to the 1755 event with large uncertainties (green stars). The orange star represents the area of the Lisbon earthquake and the filled blue star the location of the deposits in Martinique. See Data Repository for precise locations and references.





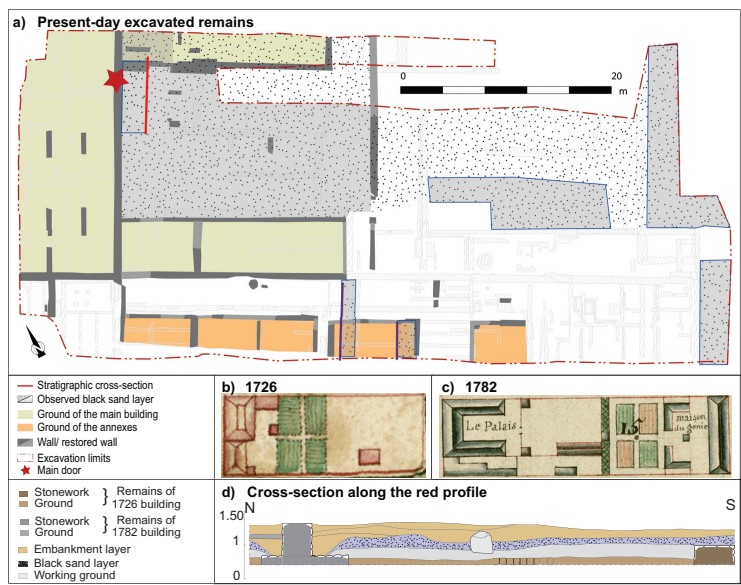

**Figure 2.** Fort-de-France site Cour d'Appel. (a) Schematic map of the excavation where the black sandy layer is represented by dots (with dark grey background when it has been reached during the excavation, and white background otherwise). (b) The same place on the 1726 land registry, with the XVII building. (c) the XVIII building in 1782. (d) Stratigraphic cross-section. Left scale is height above sea level. The black sandy layer covers the remnants of the 1726 building walls (on the right), which were already destroyed at the time when the sand was put in place, and has been cleared when the walls of the new building is built up.




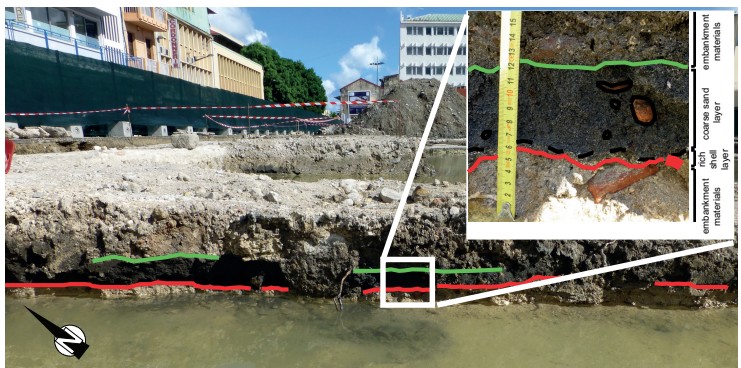

**Figure 3.** Tsunami deposit of Fort-de-France: it composed of a lower thin light grey layer delimited by a basal erosive contact of embankment materials (red line) and a upper thick dark grey layer overtopped (green line) by embankment materials. (a) View from the west of the excavation site, (b) zoom on the two sandy layers that composed the tsunami deposit: the thick black sandy layer contains shells (broken or not), gravels, brick pebbles and pieces of wood, and the light grey layer is rich in shell and coral.





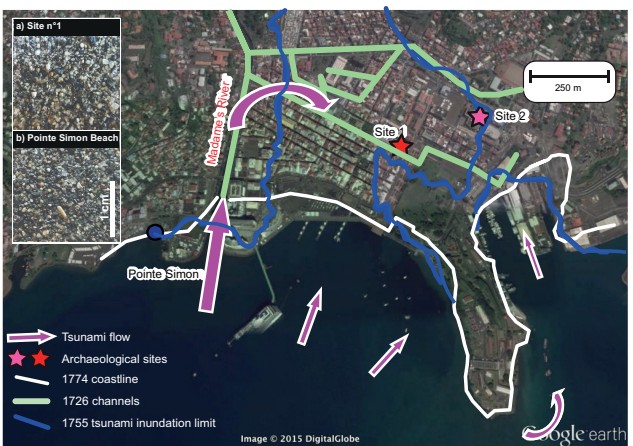

**Figure 4.** Distribution of 1755 tsunami flows entering Fort-de-France: The light grey deposit came directly with the frontal tsunami waves, whereas the thick black sandy layer was settled by the Madame River tsunami-induced bore, using the channels that no longer exists. The location of the archaeological excavations (site 1 being the Cour d'Appel and site 2 the second site) are reported and are within the inundation limit obtained using numerical modeling (from Roger et al., 2011). Left insets: (a): sand from site 1 and (b) sand from Pointe Simon beach (the blue dot on the map).