# Peer review of "Tsunami deposits in Martinique related to the 1755 Lisbon earthquake"

_Natural Hazards and Earth System Sciences, 2017_

## Referee Comment (RC1) · M. Engel (Referee) · 11 Sep 2017

The manuscript "Tsunami deposits in Martinique related to the 1755 Lisbon Earthquake" by Valérie Clouard and colleagues is dedicated to the identification of tsunami deposits in a so far largely neglected region. The Caribbean Basin is exposed to the hazard of near- and far-field tsunamis, whereas long-term recurrence patterns are very poorly known due to the short and and in some places fragmented historical record. Better constraints on the distribution of tsunami deposits is urgently needed to address regional, long-term coastal risk assessment. But also regional tsunami deposits clearly assigned to historical events are strongly required as a reference for palaeotsunami deposits.

[Figure]

Clouard et al. present a sand layer of c. 8+/-2 cm thickness with a thin basal layer of light sand and broken shells overlain by a sub-horizontally bedded layer of black coarse sand containing marine shells, small pebbles and few pieces of wood. Its chronology is highly interesting, as it is vertically confined by well dated archaeological layers, building horizons or building foundations, respectively. Deposition must have occurred between 1726 and 1783 (hard to say: while the text gives 1983 on Page 4, Line 5, Figure 2 says 1782). However, while the stage is set perfectly for a thorough study on tsunami deposits by unequivocal historical and archaeological chronology – and as much as I regret having to says this – the list of shortcomings regarding the core of the manuscript, i.e. the sedimentological analysis and interpretation, is rather long.

There is no data of the tsunami deposit's sedimentary characteristics apart from some superficial descriptions. New sites of tsunami deposits are desperately needed in the Caribbean, but they only help when thorough documentation and data are provided. In their benchmark paper on progress in palaeotsunami research, Goff et al. (2012, p. 83) state that "defining whether a deposit relates to a palaeotsunami as opposed to palaeostorm is far more complex and can only be achieved convincingly through a multidisciplinary approach using a suite of proxies." I would also extend this statement to historical deposits. In a Caribbean context, "studies using high-resolution methods of bedform and stratigraphical documentation and generating consistent chronological models with independent age control [this, however, is provided in an excellent way by the present study!], combined with refined inverse and forward models of sediment transport and deposition, are required to reconstruct long-term patterns of magnitude and frequency of palaeotsunamis in the Caribbean – a prerequisite for reliably mapping hazard-prone areas" (Engel et al., 2016, p. 290).

Unfortunately, the authors do not present efforts to validate their field-based estimation of sub-horizontal bedding by high-resolution grain size measurements. No taxonomic or taphonomic data on macro- or microfossils or geochemical or mineralogical data is given to better constrain sediment source areas and local flow patterns. No further

data on spatial extent and geometry of the deposit is given apart from the archaeo-logical trench and a diffuse outlook towards another local site. Sedimentary changes on the trench scale are not presented, as tsunami deposits in many cases often show small scale but significant changes over distances of only a few metres. Details of the stratigraphic sections are poorly visible in the photographs, sub-horizontal lamination cannot be inferred by the reader. On a trench scale, rip-up clasts should be expected if the lower contact was erosional (even though the reader does not learn about sed-imentary details of the vertically confining layers). In Chapter 3, references are made to the source area ("The thin lightly-colored layer at the basement of the deposit can be attributed to the bottom of Fort-de-France's Bay, which presently exhibits the same kind of materials."), but standards of scientific research demand a proof of these simi-larities based on sedimentological data of both deposits presented in the paper. Taking pictures of beach sand and the trench (Figure 4) definitely is not sufficient to convince the reader of the sediment source of the deposit. Also Chapter 4 contains many spec-ulations not based on sediment data. I rate the documentation of the present deposits as below standards required for international science publishing.

The most remarkable argument pro-1755 Tsunami in Chapter 2 – one that really left me puzzled – is that during the narrow time window of the possible age of the deposit, historically constraint by the age of two buildings (1726 and 1783), no major hurricanes occurred on Martinique. For some reasons, no mention is made of The Great Hurricane of 1780, the deadliest hurricane in Caribbean history with ca. 22,000 fatalities mostly on Martinique and Barbados. At the moment, I do not have access to the most detailed sources on this event (e.g. Ludlum, 1963; Depradine, 1989), but in St. Pierre, the former capital of Martinique on the slopes of Mt. Pelée, the storm surge and waves seemed to have washed away many houses and took vast numbers of lives (Neely, 2012). Not considering this iconic event in the history of Martinique falling into the time window of the deposit exposes a degree of carelessness in the discussion and interpretation of the deposit which is not complying with common standards in tsunami geoscience.

Furthermore, there are many tools available regarding inverse modelling based on high-resolution grain-size distribution in order to infer flow characteristics of a tsunami which are well accessible. Such data would be helpful to compare with existing historical accounts and the inundation model of Roger et al. (2011) in order to further substantiate the interpretation of a tsunami. Regarding literature on both regional and global tsunami deposits, some of the relevant state-of-the-art literature is not cited (see details in minor edits).

Last but not least, why have the authors not dug deeper (or took a sediment core) in order to search for predecessor deposits? The site seems to provide a good potential and there is a historical deposit to compare with.

Recommendation:

The authors have discovered a highly interesting deposit, a site with a high potential, which may be related to the 1755 tsunami. As the current state of investigations, in my opinion, does not justify international publication based on the reasons detailed above and below, I suggest to study the deposit in much more detail at Site 1 and in the new locations (Site 2 and elsewhere) by applying a multi-proxy approach. Coming up with high-resolution sedimentology and spatial extent (maybe using near-surface geophysical prospection), it could indeed represent an important reference deposit for the entire region and make a very useful future publication. I sincerely hope this review is being perceived as constructive - which is its pure intention! -, even though I am aware that it may not exactly be what the authors were hoping for.

P.S.: Apart from this rating, the manuscript would benefit from language editing, as wording is sometimes not straightforward and grammatical tenses got mixed up in some sections.

Minor edits:

L5: If considering tropical islands in general, the correct term is "tropical cyclone",

"hurricane" can only be used in a regional text

L5: "One notable exception concerns deposits sealed by subsequent events" – What does this mean? Preservation of a deposit by a subsequent (gradual/long-term or event) deposit(?) automatically enables differentiation between tsunami or storm deposit?

L6-7: Only archaeological remains? What about sedimentological evidence, which a tsunami hypothesis should be based on? The main sedimentary criteria and data supporting the tsunami hypothesis should be listed here.

L7: It is very unusual to take thickness as the main sedimentary criterion for tsunami deposition. Thickness of an event deposit is more a function of sediment availability at the source, onshore topography, surface roughness, and distance to the shoreline (and post-dpositional preservation potential) than just the hydrodynamic character of the event. Therefore, I think that the sentence "We explain the thickness of the deposit by a tsunami-induced bore in the mangrove drainage channels of Fort-de-France" is not appropriate as it does not consider the entirety of the local sedimentary and geomorphic environment into account.

P1L9: "indicate" – put another gerund here

P1L15 ff.: Hayes et al., 2014

P1L15: two large

P1L17: As I perceive the cited literature, the estimation amounts rather to M8.0–8.5 than >8.5

P1L18-19: The case of the 1690 tsunami with a seismic source near Antigua should not be dismissed here, even though the exact mechanism is not entirely clear. Some earlier sources treat 1843 as a successor of 1690 (e.g. Kelleher et al. 1973)

P1L19: contain

P1L20: 4-ft rise

P2L1-3: Delete French, English translation is sufficient.

P2L5: Better cite the much more complete O'Loughlin and Lander (2003) instead of Lander et al. (2002).

P2L9: The 1867 tsunami has also been observed at the even though no precise details are available (Reid and Taber, 1920).

P2L14-15: I do not agree: There is abundant well-constraint sedimentary evidence in southern Portugal. The Algarve is dotted with well-dated tsunami deposits of 1755 at Martinhal, Praia do Barranco, Boca do Rio, Salgados Lowland, Quateira or Carcavai, and there are much more convincing records across the border to Spain (Huelva, Donana, Valdelagrana, Barbate, Los Lances etc.), not to forget possible evidence through boulders at Cabo Trafalgar or in Morrocco. I am happy to share references, if necessary. Based on the fact that the history of very large tsunamis in this region basically only consists of the 1755 tsunami and most studies are accompanied by thorough sedimentary investigations pointing to tsunami deposition, physical dating is not so much of an issue in most cases.

P2L15: I do not agree: There is very likely sedimentary evidence of the 1755 tsunami on Anegada (Atwater et al., 2012, 2017; Watt et al., 2012, and others in the same journal issue) and on St. Thomas (Fuentes et al., 2017).

P2L19: Parsons and Geist (2009)

P2L21: Define the "studied area": The sentence before indicates that this is the Lesser Antilles, which, geographically, extend from the Spanish Virgin Islands in the north to the ABC Islands in the south. Many possible tsunami deposits have been published in this area (see Engel et al., 2016, for an exhaustive compilation). No mention has been made so far in the text that the study area under consideration is limited to Martinique or a certain part of the island.

P2L24-26: This "characteristic pattern of alternation of soil and tsunami" only occurs in very specific coastal lowland environments, therefore I would not put this statement as general as it is at the moment.

P2L26: "…whereas boulders can eventually outcrop." – Wording is a bit imprecise: Boulders shifted by tsunamis do not miraculously outcrop, but have a clear local source and their transport pathway and depositional pattern provides useful constraints on flooding processes.

P2L28: Not exhaustive. During a tropical cyclone, boulders are in most cases shifted by storm waves, not the surge. Other processes may also play a role. During Supertyphoon Haiyan, massive boulders were shifted by infragravity waves (May et al., 2015; Soria et al., 2017) or jets of extreme vertical velocities with high acceleration in cliff-edge positions (Kennedy et al., 2017).

P2L32: From a geographical point of view, when considering tropical island environments, I suggest to consider and cite preservation studies from humid tropical environments instead of temperate environments, as the range and intensity post-depositional processes significantly differ. Better refer to the observations of Nichol and Kench (2008) or Szczuciński (2012).

P2L19-L35: This overview on the formation of tsunami deposits is very general and, let's say, not really spot-on, for reasons mentioned above. The regional examples (Anegada, Guadeloupe) seem random and only cover potential coarse-clast transport by tsunami, while the study presented here covers sand in a stratigraphic sequence. The group of Anja Scheffers conducted research on most of the larger island of the Lesser Antilles island arc (Scheffers, 2006; Scheffers and Kelletat, 2006; Scheffers et al., 2005), why is only the Guadeloupe example randomly chosen? Anyway, I strongly suggest to focus on sand-sized sediment, also in the general descriptions. There has been one unsuccessful attempt to localize sandy tsunami deposits in a wetland on Basse-Terre, Guadeloupe (Morton et al., 2006), could be relevant here.

P3L1-4: The topic is very interesting and relevant, but it is somehow curious to have a potential tsunami deposit identified solely based on "historical data" (P3L2) and "archaeological remains" P1L5-6).

P3L8ff.: 17th century, check entire manuscript

P3L18-21: past tense

P3L25: includes

I stopped here for looking at minor edits.

Figures:

Fig. 2d) May be I just get it wrong, but how can the younger (i.e. 1782) stonework or ground (not sure which shade of grey....) be beneath the suggested 1755 tsunami layer? Simple clearing does not convincingly explain how the horizontal foundations of 1782 - by the way at the same level of the 1726 building - reach beneath the suggested tsunami layer.

Fig. 3, caption: Check wording

Literature:

Atwater, B.F., ten Brink, U.S., Buckley, M., Halley, R.S., Jaffe, B.E., López-Venegas, A.M., Reinhardt, E.G., Tuttle, M.P., Watt, S.,Wei, Y., 2012. Geomorphic and stratigraphic evidence for an unusual tsunami or storm a few centuries ago at Anegada, British Virgin Islands. Nat. Hazards 63, 51–84.

Atwater, B.F., ten Brink, U.S., Cescon, A.L., Feuillet, N., Fuentes, Z., Halley, R.B., Nuñez, C., Reinhardt, E.G., Roger, J.H., Sawai, Y., Spiske, M., Tuttle, M.P., Wei, Y., Weil-Accardo. J., 2017. Extreme waves in the British Virgin Islands during the last centuries before 1500 CE. Geosphere 13, 301–368.

Depradine, C. A., 1989. Pre-1900 severe hurricanes in the Caribbean. Notes compiled

for the Caribbean Meteorological Institute, St. James, Barbados.

Engel, M., Oetjen, J., May, S.M., Brückner, H., 2016. Tsunami deposits of the Caribbean – Towards an improved coastal hazard assessment. Earth Sci. Rev. 163, 260-296.

Fuentes, Z., Tuttle, M.P., Schmidt, W.E., 2017. Sand Scripts of Past Tsunamis in Coastal Ponds of St. Thomas, U.S. Virgin Islands. Seismol. Res. Lett., doi: 10.1785/0220170038.

Kelleher, J., Sykes, L., Oliver, J., 1973. Possible criteria for predicting earthquake locations and their application to major plate boundaries of the Pacific and the Caribbean. J. Geophys. Res. 78, 2547–2585.

Kennedy, A.B., Mori, N., Yasuda, T., Shimozono, T., Tomiczek, T., Donahue, A., Shimura, T., Imai, Y., 2017. Extreme block and boulder transport along a cliffed coastline (Calicoan Island, Philippines) during Super Typhoon Haiyan. Mar. Geol. 383, 65-77.

Goff, J., Chagué-Goff, C., Nichol, S., Jaffe, B., Dominey-Howes, D., 2012. Progress in palaeotsunami research. Sediment. Geol. 243–244, 70–88.

Ludlum, D. M., 1963. Early American hurricanes, 1492-1870. American Meteorological Society, Boston, 198 pp.

May, S.M., Engel, M., Brill, D., Cuadra, C., Lagmay, A.M.F., Santiago, J., Suarez, K., Reyes, M., Brückner, H., 2015b. Block and boulder transport in Eastern Samar (Philippines) during supertyphoon Haiyan. Earth Surf. Dyn. 3, 543–558.

Millas, J. C., 1968. Hurricanes of the Caribbean and adjacent regions, 1492-1800. Academy of the Arts and Sciences of the Americas, Miami, Florida, 328 pp.

Morton, R.A., Richmond, B.M., Jaffe, B.E., Gelfenbaum, G., 2006. Reconnaissance investigation of Caribbean extreme wave deposits – preliminary observations, interpre-

tations, and research directions. USGS Open-file Report 2006-1293.

Neely, W., 2012. The Great Hurricane of 1780. iUniverse.

Nichol, S.L., Kench, P.S., 2008. Sedimentology and preservation potential of carbonate sand sheets deposited by the December 2004 Indian Ocean tsunami: south Baa atoll, Maldives. Sedimentology 55, 1173–1187.

O'Loughlin, K.F., Lander, J.F., 2003. Caribbean Tsunamis – A 500-year History from 1498–1998. Kluwer, Dordrecht.

Parsons, T., Geist, E.L., 2009. Tsunami probability in the Caribbean region. Pure Appl. Geophys. 165, 2089–2116.

Reid, H.F., Taber, S., 1920. The Virgin Islands earthquakes of 1867-1868. Bull. Seismol. Soc. Am. 10, 9–30.

Roger, J., Baptista, M.A., Sahal, A., Accary, F., Allgeyer, S., Hébert, H., 2011. The transoceanic 1755 Lisbon Tsunami in Martinique, Pure Appl. Geophys. 168, 1015–1031.

Scheffers, A., 2006b. Sedimentary impacts of Holocene tsunami events from the intra-American seas and southern Europe. Z. Geomorphol. Suppl. 146, 7–37.

Scheffers, A., Kelletat, D., 2006. New evidence and datings of Holocene paleo-tsunami events in the Caribbean (Barbados, St. Martin and Anguilla). In: Mercado-Irizarry, A., Liu, P. (Eds.), Caribbean Tsunami Hazard. World Scientific, Singapore, pp. 178–202.

Scheffers, A., Scheffers, S., Kelletat, D., 2005. Paleo-tsunami relics on the southern and central Antillean Island Arc. J. Coast. Res. 21, 263–273.

Soria, J.L.A., Switzer, A.D., Pilarczyk, J.E., Tang, H., Weiss, R., Siringan, F., Manglicmot, M., Gallentes, A., Lau, A.Y.A., Lin, A.C.Y., Ling, T.K.W., 2017. Surf beat-induced overwash during Typhoon Haiyan deposited two distinct sediment assemblages on the carbonate coast of Hernani, Samar, central Philippines. Mar. Geol.,

doi: 10.1016/j.margeo.2017.08.016.

Szczuciński, W., 2012. The post-depositional changes of the onshore 2004 tsunami deposits on the Andaman Sea coast of Thailand. Nat. Hazards 60, 115–133.

Watt, S., Buckley, M., Jaffe, B.E., 2012. Inland fields of dispersed cobbles and boulders as evidence for a tsunami on Anegada, British Virgin Islands. Nat. Hazards 63, 119–131.

---

## Referee Comment (RC2) · Anonymous Referee #2 · 11 Sep 2017

Comment on
Tsunami deposits in Martinique related to the 1755 Lisbon
Earthquake

By Clouard et al.

I appreciate the reading of the manuscript as well as the effort to look for deposits of the 1st November 1755 in Martinique.
The authors identified a new sand deposit constrained by archaeological layers dated from the 18th century that the authors link to the 1755 Lisbon tsunami and use Roger's et al tsunami simulation. In my view this is a good starting point. The authors describe a single sand deposit. Are there any other layers of interest?

There are several reports of the 1755 (or linked to) in the Caribbean but to my knowledge this is not the case of Martinique
The authors constrain the formation of the deposit between 1726 and 1783. The big event in Lisbon falls perfectly in this time interval therefore we cannot exclude it. However, the authors do claim that no major storms occurred in this period. Is this true?

I believe the paper can be published if the authors should be able to:

- Do a comprehensive description of the deposit: geometry, grain size, composition – presence of organic material, shells, foraminifera, diatoms, etc.. see e.g 2010-1239 open file report to see what is needed for tsunami identification
- Try to better exclude the occurrence of a storm or hurricane
- Improve English writing

---

## Referee Comment (RC3) · Anonymous Referee #3 · 12 Sep 2017

The contribution of Clouard et al. documents an unusual sand layer found in an archaeological excavation pit on Martinique. The authors interpret the unit as evidence of the 1755 tsunami.

The 1755 Lisbon tsunami did not only cause severe flooding in Europe, but the tsunami also propagated through the Atlantic Ocean. Tsunami run up was reported from many Caribbean Islands, such as Sint Maarten (4.5 m), Antigua (3.6 m), Saba (7 m) – just to name a few (Lander et al., 2002). Reports for Martinique state that 'At Martinique, the water was reported to have withdrawn 1.6 km and returned to inundate the upper floors of houses'. Unfortunately, the contribution of Clouard et al. lacks a thorough review of existing historic reports and does for example, not even mention the quotation which I just cited.

Besides the missing information on historic reports of the 1755 tsunami in the Caribbean, there is more useful information missing in the text. For example the monthly or annual rainfall (see page 1, line 3). The authors state that the high amounts of rainfall hamper the preservation of the deposits, but give no values of the amount of precipitation. Second, speaking about preservation potential, there are a number of studies that address this topic, e.g., McAdoo et al., 2008; Nichol and Kench, 2008; Szczucinski, 2011.; Spiske et al., 2013; Andrade et al., 2014. But only the study of Bahlburg & Spiske (2015) is cited here. This shows that the authors are not aware of basic tsunami studies/publications. The same is true for the statement that 'few deposits are can be attributed' to the 1755 tsunami (p. 2, lines 14-15). This statement is not correct, as there is a number of publications that document both sand and boulders deposited by the 1755 tsunami. Most sites are in Portugal and Spain (e.g., Dawson et al., 1995; Font et al., 2010; Rodríguez-Vidal et al., 2011), and some more in Morocco and even in the Caribbean (e.g. Atwater et al., 2017). All these studies give convincing evidence supported by data, much more convincing than the pure description of Clouard et al.

Proposed tsunami unit:

The unit is up to 8 cm thick (p. 2. line 11 gives 8 cm, line 12 gives 6-9 cm). The authors state that this is an unusual thickness. I do not understand why. I would say it is an average thickness. As the thickness is not uncommon there is no need to try to explain why it is unusually thick.

The unit includes marine shells and pebbles (p. 3, line 25): are the shells broken or articulated? Which species? Which water depth? What is the size of the pebbles?

There is only one photo of the sand unit which is not informative because it is too small. Second, you need to show details. The outcrop is several meters long, but only a section which is a few tens of centimeters wide is described. You need to use the full extent of the outcrop, as otherwise you will not get any information on spatial trends.

[Figure]

Sediment sources: p. 4, line 3 states that there is a 'distinct origin' of the material of which the sand unit is composed. However no analyses were conducted to prove the marine origin of the white sand and the fluvial origin of the black layer. An inserted small photo (Fig. 4) with terrible resolution is not enough for a comparison. Any potential source needs to be analyzed the same way (components, grain size) as the proposed event unit.

p. 4, line 21 'The thin lightly-colored layer at the basement of the deposit can be attributed to the bottom of Fort-de-France's Bay, which presently exhibits the same kind of materials.' What is the water depth from which the material is supposed to have been eroded? If you know that the seafloor has the same sediments, I assume that you had a look and took samples. These samples need to be analyzed and documented as references samples.

Methods:

This work is highly incomplete and far from a scientific study. You can easily tell with by the fact that there is no methods-chapter. There is an utmost need to include grain size analysis, microorganisms, sediment composition and other suitable methods. Are there any sorting trends? Does the unit thin in seaward or landward direction?

As stated before, the material of all potential sources needs to be analyzed, too – otherwise no comparison is possible!

The material is siliciclastic and therefore you could check its age using OSL-dating.

It is ridiculous that you state (p. 6, lines 18-20) 'Our results indicate that the use of all available geological methodologies and collaborative studies with historians and archeologists might enable us to improve our historical tsunami catalogs in the future, thus helping the preparedness of tsunami hazard plans for coastal communities' as you have not applied any of the available geological methods!

Context:

There is no information on the height above sea level of the excavation site and the event unit. How far is the site from the former coastline? Fig. 4 seems to show that the site is actually within one of the channels – is that true?

Unfortunately, the contribution shows that the authors have only very limited knowledge and understanding of the state-of-the-art in sedimentological tsunami research, sediment transport processes and tsunami flow dynamics.

p. 6, line 6 and 15 'paleotsunami' – this is not the correct term. Paleotsunami units are lithified. Thus historical tsunami deposit would be the correct term.

p. 4, lines 1-2 'and the sand particles sub-horizontal lamination (Fig. 3c) shows that the water enters the site and the sediments deposit slowly, and that there is no contribution from upper backflow.' This hardy makes sense. If the sediment particles slowly settle from the flow, normal (suspension) grading will be produced. Lamination is not a sedimentary structure that represents slow deposition! Instead lamination is caused by changes in the supply of sediment, most of all in terms of grain size and grain density.

p. 3, line 32 'The black layer globally follows the light grey layer, indicating a coeval deposition.' The deposition is not coeval, but successive otherwise one layer would not lie on top of the other.

Attribution to the 1755 event: It is stated that the event unit was deposited in between 1726 and 1783 and thus I may resemble evidence of the 1755 tsunami. However, there was another tsunami in 1767 (see chapter 3). The 1767 tsunami was a regional event and thus could have had an even greater impact than the 1755 far field tsunami. In addition, a storm event is excluded, even though large storms/hurricanes are frequently hitting the region. There was a severe hurricane in 1780 which killed thousands of people on Martinique and I do not see any argument why the described event unit could not be likewise related to this event.

There is hardly any discussion on an alternative process that could have deposited the

[Figure]

unit. Only on page 3, lines 27-30, storms are quickly mentioned and excluded without a real discussion.

Fig. 4: if the site is in or close to one of the historical channels, the settling needs to be interpreted differently. Backwash will use preexisting channels or topographic lows and you need to discuss the deposition of the unit in terms of this setting.

Thickness and event magnitude (p- 4. Lines 16-19): The thickness of a tsunami deposit is directly related to the amount and grain size of available material, in addition to the topography (thicker units in topographic lows). There is no way to relate the magnitude of inundation or even of the event to sediment thickness! p. 6, line 11: I do not understand why it is concluded that the ∼8 cm thick deposit needs to be produced by a 1 m tsunami wave. Where does this value come from? And still 8 cm is far from being unusual. The authors state (p. 6, lines 14-15) 'This indicates that the tsunami deposit thickness used in tsunami modeling is a parameter 15 that must be carefully checked in order to avoid overestimation of paleo-tsunamis and to correctly assess the tsunami hazard.' This is no new finding, but was already stated in several publications (e.g. Spiske et al., 2013). I do not see why the authors start the discussion of sediment thickness and inverse modelling because they did not do any inverse modelling. Roger et al. (2011) did forward modelling, but these results would only support the results of an inverse modelling approach.

p. 6, lines 13-14: What exactly, in your opinion, is the difference of the tsunami wave front and a tsunami-induced bore? Explain what the differences in flow are and how they influence the mode of transport and deposition. What about the role of successive waves of the tsunami wave train?

p. 6, lines 3-4 '1755 tsunami deposits can also be found in locations in eastern coastal American cities' – where should that be? Do you think that for example the US east coast was hit by the 1755 tsunami?

Language:

[Figure]

The language is in many parts very poor. Many sentences are difficult or not to understand. Just to give a few examples: p. 2, line 26 'whereas boulders can eventually outcrop' p. 3, line 31 'the black layer globally follows' p. 4, line 2 'upper backflow' – what is an upper backflow? Never heard of this before. p. 6, line 11 'the Americans' – totally unspecific. Where should that be? Fig. 1 'large uncertainties' – which uncertainties? I cannot understand to what this is related (runup height, locations, ?)

Figures and Table:

Fig. 1: runup height is given only as a number without unit. Add 'meters'

Fig. 2: add arrows that indicate the direction of the sea and the river. 'white background otherwise' – what does this mean? Are these areas interpolated? The legend only lists the black sand layer, but you state in the text that the tsunami unit is composed of a white and black layer. Add either the white layer to the map and legend or change 'black sand' into 'event unit' or 'proposed tsunami unit'

Fig. 3: add arrows that indicate the direction of the sea and the river. Sediment photo needs to be much larger and resolution needs to be much higher. Lamination is described in the text and you need to prove this in the photo!

Fig. 4: 4a has no scale! 4a and 4b need to be much larger. Again, just a visual comparison is worth nothing. Add more photos of the white layer and the proposed marine source material. Add lines for the topography. Otherwise the course of the inundation limit is hard to understand.

Tab. 1: What means 'H'? 'obs.' means 'observed'? Add explanation of the abbreviations to the captions.

Unfortunately I have to conclude that the only value of this contribution is the fact that the authors document an unusual sand layer within the remains of a historic building and its garden. The study has no scientific value as it does not meet the standards of thorough sedimentological research. The authors describe the sand unit, but do not

include any analyses. Hence, there is no data that allows for the interpretation and understanding of related processes, such as sediment entrainment, sediment transport, deposition and tsunami hydrodynamics. Yes, the unit could likely be a deposit of the 1755 tsunami. But without data and without a discussion on storm versus tsunami, it remains pure speculation.

References: Andrade, V.,Rajendran, K., Rajendran, C.P., 2014: Sheltered coastal environments as archives of paleo-tsunami deposits: Observations from the 2004 Indian Ocean tsunami. Journal of Asian Earth Sciences, 95, 331-341 Atwater, B.F., ten Brink, U.S., Cescon, A.L., Feuillet, N., Fuentes, Z., Halley, R.B., Nuñez, C., Reinhardt, E.G., Roger, J.H., Sawai, Y., Spiske, M., Tuttle, M.P., Wei, Y., Weil-Accardo, J., 2017: Extreme waves in the British Virgin Islands during the last centuries before 1500 CE. Geosphere, v. 13 (2), 301-368. Bahlburg, H. and Spiske, M., 2015: Styles of early diagenesis and the preservation potential of onshore tsunami deposits - a resurvey of Isla Mocha, Central Chile, two years after the February 27, 2010 Maule tsunami. Sedimentary Geology, 326, 33-44. Dawson et al., 1995: Tsunami sedimentation associated with the Lisbon earthquake of 1 November AD 1755: Boca do Rio, Algarve, Portugal. The Holocene, 5, 209-215. Font et al., 2010: Identification of tsunami-induced deposits using numerical modeling and rock magnetism techniques: A study case of the 1755 Lisbon tsunami in Algarve, Portugal. Physics of the Earth and Planetary Interiors 182 (2010) 187–198 Krien, Y., Dudon, B., Roger, J., and Zahibo, N., 2015: Probabilistic hurricane-induced storm surge hazard assessment in Guadeloupe, Lesser Antilles, Nat. Hazards Earth Syst. Sci., 15, 1711-1720, https://doi.org/10.5194/nhess-15-1711-2015. Krien, Y., Dudon, B., Roger, J., Arnaud, G., and Zahibo, N., 2017: Assessing storm surge hazard and impact of sea level rise in Lesser Antilles-Case study of Martinique, Nat. Hazards Earth Syst. Sci. Discuss., https://doi.org/10.5194/nhess-2017-148. Lander, J.F., Whiteside, L.S., and Lockridge, P.A., 2002, A brief history of tsunamis in the Caribbean Sea: Science of Tsunami Hazards, v. 20, p. 57–94. McAdoo, B.G., Fritz, H.M., Jackson, K.L., Kalligeris, N., Kruger, J., Bonte-Grapentin, M., Moore, A.L., Rafiau, W.B., Billx, D., 2008. Solomon Islands tsunami, one year later. Eos 89, 169–170. Nichol,

S.L., Kench, P.S., 2008. Sedimentology and preservation potential of carbonate sand sheets deposited by the December 2004 Indian Ocean tsunami: South Baa Atoll, Maldives. Sedimentology 55, 1173–1187. Rodríguez-Vidal et al. 2011: The recorded evidence of AD 1755 Atlantic tsunami on the Gibraltar coast. Journal of Iberian Geology 37 (2) 2011: 177-193 Spiske, M., Piepenbreier, J., Benavente, C., Bahlburg, H., 2013: Preservation potential of tsunami deposits on arid siliciclastic coasts. Earth-Science Reviews, 126, 58-73. Szczucinski, W., 2011. The post-depositional changes of the onshore 2004 tsunami deposits on the Andaman Sea coast of Thailand. Natural Hazards 60, 115–133. http://www.hurricanescience.org/history/storms/pre1900s/1780/

---

## Author Comment (AC1) · 19 Feb 2018

General reply to the three referees

The reviews from the three referees share a common approach of what must be the data associated with the presentation of a new tsunami deposit. In our mind, our paper was the description of an important overwash deposit in Martinique, FWI, that we managed to relate to a tsunami event thanks to archaeological and geomorphic analysis: our goal was not to lead a sedimentary study of this deposit. However, we understand that our result would be more useful to the tsunami community with sedimentary data. The sedimentary analysis is now under process and some results are shown below. If possible, we could add our colleague sedimentologist as a co-author. We have also

noticed that, in general, our archaeological and geomorphic analysis should be re-fined, in addition with a detailed description of Martinique climate and Fort-de-France topography.

We report below in detail the responses to the remarks of referee #1.

Reply to M. Engel, referee #1 :

P C2, §1 : the sedimentological analysis and interpretation

When we received this review, we sent for analyzing samples from our excavation site 1 (Court of Appeal) and from the river mouth to get sedimentary information. Later, we sent sample from Fort-de-france Bay and our colleagues in charge of excavation site 2, the Police building, sent samples to the same laboratory. We have got the analyzes of the first samples, and those from site 2 and from Fort-de-france Bay will be done for the end of February. We now have the grain size distribution (Figure 1) and other parameters (mean and median size, variance, skewness, kurtosis, etc). We'll add this information in our paper.

These analyses also include compositional data with: Ba, Sb, Sn, Cd, Pd, Ag, Mo, Zr, Sr, Rb, As, Se, Au, Pb, W, Zn, Cu, Ni, Co, Fe, Mn, Cr, V, Ti, Ca, K, Al, P, Si, Cl, S, Mg, $SiO_2$, $MgO$, $Al_2O_3$, $P_2O_5$, $K_2O$, $CaO$, $TiO_2$, $MnO$, $Fe_2O_3$, U, Th, Hg, Sc, Cs, Te. We'll report it in our paper.

P C2, §2: "The most remarkable argument pro-1755 Tsunami in Chapter 2 – one that really left me puzzled – is that during the narrow time window of the possible age of the deposit, historically constraint by the age of two buildings (1726 and 1783), no major hurricanes occurred on Martinique. For some reasons, no mention is made of The Great Hurricane of 1780, the deadliest hurricane in Caribbean history with ca. 22,000 fatalities mostly on Martinique and Barbados."

We do not ignore the major hurricane of 1780, which devastated all the Lesser Antilles and a part of the Greater Antilles, from St Vincent to Jamaica. The analysis of historical

reports from the construction indicates that the Court of Appeal was built before 1773. In addition, the results from site 2 archaeological analysis are now available (Navetat, Nadeau et al, 2016): The construction of the Police Station building began after 1761 and not later than 1770. In site 2, the deposit layer is everywhere above the mangrove and just below the first embankments. This clearly indicates that the deposit predates 1770. We note that our demonstration is poorly written. We'll make appear clearly in our reviewed paper that the observed deposits at site 1 and site 2 predate the 1780 hurricane, although the cartographic documents that we used give an age bracket between 1726 and 1782.

P C4 §2: "Last but not least, why have the authors not dug deeper (or took a sediment core) in order to search for predecessor deposits? The site seems to provide a good potential and there is a historical deposit to compare with."

We agree with referee #1 that interesting results could have been achieved by coring the area. Unfortunately, it has not been possible to schedule such a coring within the very short period of the 3-week excavation: one must remember that the main objective of these excavations were to better understand Fort-de-France first settlement and to improve our knowledge of the successive construction stages. When we began to work on the overwash deposit layer at site 1, it was the end of the excavation, as it corresponds to the lower depths reached: there was no more time to get the only core drill of the island, and impossible to get a core drill from another country.

P C4 Minor edits: L5: "tropical cyclone" vs "hurricane"

We'll check in our paper, at each occurrence of the terms "cyclone" or "hurricane", the appropriate word. However, going to NOAA website (https://oceanservice.noaa.gov/facts/cyclone.html), we found these definitions that we assume to be correct: "Once a tropical cyclone reaches maximum sustained winds of 74 miles per hour or higher, it is then classified as a hurricane, typhoon, or cyclone depending upon where the storm originates in the world" and "Hurricanes, cyclones,

and typhoons are all the same weather phenomenon; we just use different names for these storms in different places. In the Atlantic and Northeast Pacific, the term "hurricane" is used. The same type of disturbance in the Northwest Pacific is called a "typhoon" and "cyclones" occur in the South Pacific and Indian Ocean."

P C5 Minor edits: L5: "One notable exception concerns deposits sealed by subsequent events" – What does this mean?

We wanted to speak about a subsequent event for which dating is possible with precision, which provides an age bracket for the overwash. We'll change this sentence.

P C5 Minor edits: L6-7: Only archaeological remains? What about sedimentological evidence, which a tsunami hypothesis should be based on? The main sedimentary criteria and data supporting the tsunami hypothesis should be listed here.

In the initially submitted paper, our sedimentary analysis was rough and it is why we do not mention it in the abstract. We'll change it in the revised version, adding the analysis done during this review process.

P C5 Minor edits: L7: "...Therefore, I think that the sentence "We explain the thickness of the deposit by a tsunami-induced bore in the mangrove drainage channels of Fort-de-France" is not appropriate as it does not consider the entirety of the local sedimentary and geomorphic environment into account."

We agree with referee #1 that the way we reached this conclusion can be improved by a better description of the local geomorphic and sedimentary context, which is not well described in our paper. We'll add a paragraph in the revised paper.

P C5 Minor edits: P1L15 ff.: Hayes et al., 2014

The reference is correct. It is: "Geophys. J. Int. (2013) doi: 10.1093/gji/ggt385, Quantifying potential earthquake and tsunami hazard in the Lesser Antilles subduction zone of the Caribbean region, Gavin P. Hayes, Daniel E. McNamara, Lily Seidman and Jean Roger, Accepted 2013 September 20. Received 2013 September 17; in original form

2013 June 20.

P C5 Minor edits: P1L17: As I perceive the cited literature, the estimation amounts rather to M8.0–8.5 than >8.5

Yes, it is right. We wanted to write >8.3, and a conservative magnitude should be 8. We'll change it to "M8.0–8.5".

P C5 Minor edits: P1L18-19: The case of the 1690 tsunami with a seismic source near Antigua should not be dismissed here, even though the exact mechanism is not entirely clear.

About 1690 earthquake, from Bernard and Lambert (1988): "Recently, Feuillard (1984) showed that the intensity IX reported by Robson (1964) in Guadeloupe for the 1690 earthquake on the basis of a British document is very likely to have concerned the islands of Ste. Eustache and St. Christophe, which were French at this moment, and not the French island of Guadeloupe. The few documents that Feuillard could find in Guadeloupe suggest a more likely intensity of VI on this island. This moves the source area towards the North, and considerably reduces its extension."

In Feuillard (1984), there are several French documents to justify this assumption. It is also reported in Feuillet et al (2011), where it is proposed that it ruptured the en echelon fault system to the west of the volcanic arc. Our paper is indeed not the place to discuss this specific event, but according to these analysis, it is doubtful that 1690 earthquake was a thrust event. In addition, the tsunami is described in Feuillet et al. (2011): "This earthquake probably triggered a tsunami, since it was reported that the sea withdrew over a distance of 200 m in Charleston (western part of Nevis) and returned after 2 min." This 2-mn return period could be more easily associated with a local landslide. We'll try to better take it into account in our paper.

P C6 Minor edits: P2L5: Better cite the much more complete O'Loughlin and Lander (2003) instead of Lander et al. (2002).

[Figure]

We used the Science of Tsunami Hazards article as it was published prior to O'Loughlin and Lander's book and easiest to get. We'll also cite O'Loughlin and Lander's book.

P C6 Minor edits: P2L9: The 1867 tsunami has also been observed at the even though no precise details are available (Reid and Taber, 1920).

We'll add the 1867 tsunami in our list of historical tsunami, line 28 of our paper although Reid and Taber (1920) have written "The waves were also noted at Martinique, but we have no description of them."

P C6 Minor edits: P2L14-15: I do not agree: There is abundant well-constraint sedimentary evidence in southern Portugal…

Our phrasing is clumsy as we wanted to highlight the lack of serious evidence of deposits in North and South America and in the Caribbean. Information on 1755 deposit does exist in Europe and we reported it Table 1, by site. If we forgot references, we apologize. What we wanted to underline is that 1755 tsunami impact evidence on the western side of the Atlantic Ocean is mostly based on historical records. Recently, B. Atwater et al. (2017) proposed in a conservative way that it could be present in Anegada : "More extensive overwash, perhaps by the 1755 Lisbon tsunami, is marked primarily by a sheet of sand and shells found mainly below sea level beneath the floors of modern salt ponds. This sheet extends more than 1 km southward from the north shore and dates to the interval 1650–1800 cal yr CE." We'll change our phrasing in the revised paper.

P C6 Minor edits: P2L15: I do not agree: There is very likely sedimentary evidence of the 1755 tsunami on Anegada (Atwater et al., 2012, 2017; Watt et al., 2012, and others in the same journal issue) and on St. Thomas (Fuentes et al., 2017).

We know Atwater et al. (2012, 2017) and Watt et al. (2012) articles. The paper from Fuentes et al (2017) was not yet published when our paper was submitted. One can note that in NOAA database, Fuentes et al (2017) is now (there was none when we

submitted) the only reference to 1755 in the Americas. We'll add this reference to our revised version.

P C6 Minor edits: P2L19: Parsons and Geist (2009)

2008 will be corrected to 2009.

P C6 Minor edits: P2L21: Define the "studied area": The sentence before indicates that this is the Lesser Antilles, which, geographically, extend from the Spanish Virgin Islands in the north to the ABC Islands in the south.

The term "Lesser Antilles" is indeed a misnomer in our text, although often used in the Antilles. We meant "the islands of the recent volcanic arc of the Lesser Antilles". We'll change that.

P C7 Minor edits: P2L24-26: This "characteristic pattern of alternation of soil and tsunami . . . whereas boulders can eventually outcrop" only occurs in very specific coastal lowland environments, therefore I would not put this statement as general as it is at the moment.

We'll change this sentence, too short to describe in general all kind of tsunami deposit.

P C7 Minor edits: P2L32: From a geographical point of view, when considering tropical island environments, I suggest to consider and cite preservation studies from humid tropical environments instead of temperate environments, as the range and intensity post-depositional processes significantly differ. Better refer to the observations of Nichol and Kench (2008) or Szczucinski (2012).

We agree to use examples from similar islands. However, the study from Nichol and Kench (2008) takes place in the Maldives archipelago, composed of atolls, which environment is not the same than our elevated islands. We'll refer to Szczucinski's (2012) study.

2017-238, 2017.

**Mean grain size distribution**

Legend:
- Pointe Simon: River mouth
- Court of Appeal

Zone de traçage

**Fig. 1.** Mean grain-size ($\mu$m) distributions of the samples

---

## Author Comment (AC2) · 19 Feb 2018

General reply to the three referees

The reviews from the three referees share a common approach of what must be the data associated with the presentation of a new tsunami deposit. In our mind, our paper was the description of an important overwash deposit in Martinique, FWI, that we managed to relate to a tsunami event thanks to archaeological and geomorphic analysis: our goal was not to lead a sedimentary study of this deposit. However, we understand that our result would be more useful to the tsunami community with sedimentary data. The sedimentary analysis is now under process and some results are shown below. If possible, we could add our colleague sedimentologist as a co-author. We have also

noticed that, in general, our archaeological and geomorphic analysis should be refined, in addition with a detailed description of Martinique climate and Fort-de-France topography.

We report below in detail the responses to the remarks of referee #2.

Reply to anonymous referee #2 :

- Do a comprehensive description of the deposit: geometry, grain size, composition – presence of organic material, shells, foraminifera, diatoms, etc.. see e.g 2010-1239 open file report to see what is needed for tsunami identification

When we received this review, we sent for analyzing samples from our excavation site (Court of Appeal) and from the river mouth to get sedimentary information. Later, we sent sample from Fort-de-france Bay and our colleagues in charge of excavation site 2, the Police building, sent samples to the same laboratory. The laboratory had time to make the analyzes of the first samples, and they will do those from site 2 and from Fort-de-france Bay for the end of February. We now have the grain size distribution (Figure 1) and other parameters (mean and median size, variance, skewness, kurtosis, etc). We'll add this information in our paper.

These analyses also include compositional data with: Ba, Sb, Sn, Cd, Pd, Ag, Mo, Zr, Sr, Rb, As, Se, Au, Pb, W, Zn, Cu, Ni, Co, Fe, Mn, Cr, V, Ti, Ca, K, Al, P, Si, Cl, S, Mg, $SiO_2$, MgO, $Al_2O_3$, $P_2O_5$, $K_2O$, CaO, $TiO_2$, MnO, $Fe_2O_3$, U, Th, Hg, Sc, Cs, Te. We'll report it in our paper.

- Try to better exclude the occurrence of a storm or hurricane

Chronology at site 1: The analysis of historical reports from the construction indicates that the Court of Appeal was built before 1774. We found this information in Marion (2000): in 1763, M. Daux, the owner of the building, is nominated tax collector by the king. The extension of the building is done around 1770. It is during this construction period that the overwash occurred. In 1774, following troubles with the law, his goods

are seized by the king and his residence in Fort-Royal is transformed in Courthouse. Chronology at site 2 : the archaeological report from site 2 is now available (Navetat, Nadeau et al, 2016): The construction of the Police Station building began after 1761 and not later than 1770. In site 2, the deposit layer is everywhere above the mangrove and just below the first embankments. It is one more evidence to say that the deposit predates 1770 and even 1761.

Concerning the occurrence of hurricane, we used the review from Romer (1932). He gives a detailed list of the historical stormy events in Martinique between 1635 and 1932. Between 1726 and 1782, there is 11 storms or cyclonic phenomena: 1/10/1753 (correctly described: damages on small boats); 12/09/1756 (correctly described: damage in the East and South of the island); 12/09/1758 (poorly described, just: wind gusts); 7/11/1760 (shortly described: 12 boats to the coast in St Pierre); 09/1965 (poorly described, just: hurricane in Martinique, Guadeloupe and St Christophe); 13/08/1766 (devastating hurricane over Martinique, 410 casualties, 80 boats lost); 1775: hurricanes the 30th of July and the 25th of August (no description); 5/09/1776 (shortly described: 22 boats to the coast); 3/10/1779 (shortly described: strong gale, the only damages occurred on-land); and 12/10/1780 (correctly described). 1780 major hurricane devastated all the Lesser Antilles and a part of the Greater Antilles, from St Vincent to Jamaica. In Martinique, a tidal surge threw the boats to the coast, and in Fort-Royal, the cathedral and 140 houses were overturned, there was 7000 casualties.

To summarize, the historical and archaeological information does not enable to get a more precise construction time than before 1770. And before 1770, there is just one noticeable hurricane in 1766 probably not strong enough to devastate Fort-Royal, and three tsunamis, 1761, 1767 and 1755.

However, we note that our demonstration is poorly written. We'll make appear clearly in our reviewed paper that the observed deposits at site 1 and site 2 predate the 1780 hurricane, although the cartographic documents that we used give an age bracket between 1726 and 1782, which can be confusing.

- Improve English writing

We apologize for our English. A preliminary version was reviewed by an English colleague, but later, a lot of modifications were done. We'll ask to a colleague whose mother tongue is English to review our final version.

References: Marion, G. G. (2000) : Administration des finances en Martinique : 1679-1790, L'Harmattan Ed. Romer, J.: Liste chronologique des cyclones à la Martinique (1635 à 1932), Tech. rep., Service Météorologique et de Physique du Globe, 1932.
* * *
[Figure]

**Mean grain size distribution**

**Fig. 1.** Mean grain-size ($\mu$m) distributions of the samples

---

## Author Comment (AC3) · 19 Feb 2018

General reply to the three referees to

"Tsunami deposits in Martinique related to the 1755 Lisbon earthquake" by Valérie Clouard, Jean Roger, and Emmanuel Moizan

The reviews from the three referees share a common approach of what must be the data associated with the presentation of a new tsunami deposit. In our mind, our paper was the description of an important overwash deposit in Martinique, FWI, that we managed to relate to a tsunami event thanks to archaeological and geomorphic analysis: our goal was not to lead a sedimentary study of this deposit. However, we understand that our result would be more useful to the tsunami community with sedimentary data.

The sedimentary analysis is now under process and some results are shown below. If possible, we could add our colleague sedimentologist as a co-author. We have also noticed that, in general, our archaeological and geomorphic analysis should be refined, in addition with a detailed description of Martinique climate and Fort-de-France topography.

We report below in detail the responses to the remarks of referee #3.

Reply to anonymous referee #3 :

P. C1 : "Tsunami run up was reported from many Caribbean Islands, such as Sint Maarten (4.5 m), Antigua (3.6 m), Saba (7 m) – just to name a few (Lander et al., 2002)."

We report the historical records and the deposits related to 1755 tsunami in the table 1 of the electronic supplement. We don't know if referee #3 had an access to this supplementary material. Figure 1 is a snapshot of the Caribbean part of Table 1.

P. C1 : Reports for Martinique state that 'At Martinique, the water was reported to have withdrawn 1.6 km and returned to inundate the upper floors of houses'.

We deliberately don't cite this information. We know this sentence from Lander (p.16, 1996) and also in Lander et al. (2002). Among the various references which appear in his articles, none refers to this observation, and we were not able to go back to the source of the initial information.

Conversely, when looking at NOAA database (Figure 2), it appears information for Martinique but none about any withdrawn. The last line in the NOAA databse extract is the only inundation information. The coordinates are those of St Pierre with a 1.8m wave height and a 1500m inundation distance. It is also the only place where one house was damaged. When looking at Roger et al (2011), one of us, it appears for Martinique the information of Figure 3 (with the references included)

Finally, we never found any information about this important withdraw, and we prefer

not to include it.

As supplementary material is not always read, we propose to reviewer #3 to include our table in the paper.

P. C2, first §: "monthly or annual rainfall" :

We'll add this information, as well as a better geomorphic description of the area.

P. C2, first §: "Second, speaking about preservation potential, there are a number of studies that address this topic, e.g., McAdoo et al., 2008; Nichol and Kench, 2008; Szczucinski, 2011.; Spiske et al., 2013; Andrade et al., 2014. But only the study of Bahlburg & Spiske (2015) is cited here."

McAdoo et al's (2008) report an interesting testimony on Solomon Islands earthquake and tsunami, where the geologists arrived in the affected area 1 month after the event. In their conclusion, they said "however, the longer-term ecologic and economic impacts remain to be seen." Why would we have to cite this report?

Nichol and Kench's (2008) paper also stated that "the preservation potential of these tsunami deposits is low to moderate" in the Maldives archipelago, composed of atolls and visited 2 years after the 2004 Indian ocean tsunami. They compare deposits in several atolls for which the distance to the source is ca 2500 km with 1.5 to 2.5-m wave height, to compare with the 5600 km between Lisbon and Martinique and our historical observation of 1-m height in Fort-de-France, 260 years ago. We could add this reference to our paragraph on the difficulty to get preserved deposits in tropical elevated island, although all the parameters are not comparable.

Spiske et al.'s (2013) went in Peru to study specifically the preservation of tsunami in arid coasts. We are not sure that their results are relevant in our Caribbean context of wet tropical islands.

Szczucinski's (2011) paper describe the evolution of the Indian Ocean 2004 tsunami deposit during the 5 years following the flooding under conditions of tropical climate

with high seasonal rainfall. He concludes by: "tsunami deposits that are thinner than 10 cm have little preservation potential. Consequently, the sedimentary record of tsunamis with a run-up smaller than 3 m is not likely to be preserved at all" and "Any modelling of paleotsunamis from their deposits must take into account post-depositional changes." We'll add this reference in our discussion, to highlight the positive effect of human settlement to preserve a deposit layer due to a less than 3-m tsunami, and the results than could be obtained if archaeological excavations close to the shore were systematically analyzed in terms of tsunami deposits.

Through the observation of 2004 tsunami, Andrade et al. (2014) try to identify conducive coastal environments to tsunami deposit preservation, in areas also subject to storms. We can include this reference in our introduction, along with the other papers more specifically related to the Caribbean context and already cited on this topic: Spiske et al., 2008, Atwater et al., 2012; Buckley et al., 2012, Scheffers et al., 2005...

P. C2, first §: "The same is true for the statement that 'few deposits can be attributed' to the 1755 tsunami (p. 2, lines 14-15). This statement is not correct, as there is a number of publications that document both sand and boulders deposited by the 1755 tsunami. Most sites are in Portugal and Spain (e.g., Dawson et al., 1995; Font et al., 2010; Rodríguez-Vidal et al., 2011), and some more in Morocco and even in the Caribbean (e.g. Atwater et al., 2017)."

Deposits related to 1755 tsunami are listed in the electronic supplement (Table 1 of the article). We don't know if referee #3 had an access to this supplementary material (see Figure 3). Atwater et al. (2017) is indicated for Anegada. However, it can be noted that Atwater et al. (2017) present their results in a very conservative way: " (for the 1650–1800 event,) This reasoning leaves either a tsunami of nearby origin or an unusual storm that produced tsunami-like bores as the cause of the 1200–1480 catastrophe." Furthermore, this result is not included in the NOAA database. It is probably not our duty to conclude that their observations do correspond to 1755 tsunami if the authors do not write it.

Dawson et al (1995) and Hinson et al (1996) are indicated for Portugal, Boca do Rio, Algarve deposit. Font et al, in 2010, resampled the same Algarve deposit and used numerical modeling and rock magnetism techniques to confirm the results from Dawson et al. (1995) and Hinson et al (1996). As the main goal of Font et al.'s paper was to propose a new method to characterize on-land deposits related to strong overwash events and to relate them to tsunami because the mixture with underlying layers indicate a high energy phenomena, we have considered that it was not useful to add this reference in our table 1. We'll add an entry in our table 1 for Gibraltar deposit described in Rodríguez-Vidal et al., 2011. We apologize for not including this reference in our table 1.

P C2: "Proposed tsunami unit: The outcrop is several meters long, but only a section which is a few tens of centimeters wide is described. You need to use the full extent of the outcrop, as otherwise you will not get any information on spatial trends."

The full extent of the outcrop is drawn in our paper on Fig 2. In grey is represented the horizontal extension of the deposits in the whole area. As it can be seen Figure 4, the archaeological excavation did not reach the deposit layer everywhere.

The first reason is that this was an archaeological site, where the main goal was to better understand the first colonial settlement in Fort-de-France, not to characterize tsunami deposit and the site was dug only to the anthropogenic layers, which were above the deposit layer. At some places, deeper excavation was done only to test the possibility of pre-colombian remains.

The second reason is the depth of the deposit layer corresponds to the groundwater level (Figure 5). Figure 6 shows the water pumps working at site 2. Thus, it was not possible to dig everywhere at this depth.

P C2: "Proposed tsunami unit: The unit is up to 8 cm thick (p. 2. line 11 gives 8 cm, line 12 gives 6-9 cm).

No, we have written " In direct contact with the archaeological levels associated with this new construction, an 8 cm-thick sandy layer is present. It consists of a 1 cm-thick rich shelly lighter-colored layer at its base and an upper 6-9 cm-thick black layer". Small variations obviously exist in the thickness (see Figure 7).

We'll add the term "average" thickness in the first part of our sentence. Thank you for noting.

P C2: The authors state that this is an unusual thickness. I do not understand why. I would say it is an average thickness. As the thickness is not uncommon there is no need to try to explain why it is unusually thick. "

We'll try to better explain why this average thickness of 8 cm is not common for a teletsunami of 1 meter. The thickness of the deposit depends on many factors such as the quantity of available material, the topography (including in our case the geometry and orientation of the construction), the inundation processes (wave speed and number of waves), the occurrence or not of backflow (not in our case), and the height of the waves.

p. C3, §1: "Sediment sources: p. 4, line 3 states that there is a 'distinct origin' of the material of which the sand unit is composed. However no analyses were conducted to prove the marine origin of the white sand and the fluvial origin of the black layer. An inserted small photo (Fig. 4) with terrible resolution is not enough for a comparison. Any potential source needs to be analyzed the same way (components, grain size) as the proposed event unit. Âż

We'll add the sediment analysis and better photos, like Figure 7 and Figure 8.

P C3: p. 4, line 21 'The thin lightly-colored layer at the basement of the deposit can be attributed to the bottom of Fort-de-France's Bay, which presently exhibits the same kind of materials.' What is the water depth from which the material is supposed to have been eroded? If you know that the seafloor has the same sediments, I assume that you

had a look and took samples. These samples need to be analyzed and documented as references samples.

We have sampled Fort-de-France bay in shallow waters (200m from the shore, 5m depth) and the sand of the beach, which is rougher and contains more shells than the sand of the bay. However, we will not try to convince anyone that 21th century environmental conditions are the same than those of the mid-XVIII century for this bay and beach in the nowadays devastated, bare and contaminated bay of Fort-de-France. In addition, historical nautical charts indicate a now disappeared fringing reef, just in front offshore our site.

However, sediment analysis of the bay and beach of Fort-de-France samples will be included in our revised manuscript, with the grain size distribution (Figure 9) and other parameters (mean and median size, variance, skewness, kurtosis, etc). We'll add this information in our paper.

These analyses also include compositional data with: Ba, Sb, Sn, Cd, Pd, Ag, Mo, Zr, Sr, Rb, As, Se, Au, Pb, W, Zn, Cu, Ni, Co, Fe, Mn, Cr, V, Ti, Ca, K, Al, P, Si, Cl, S, Mg, $SiO_2$, MgO, $Al_2O_3$, $P_2O_5$, $K_2O$, CaO, $TiO_2$, MnO, $Fe_2O_3$, U, Th, Hg, Sc, Cs, Te. We'll report it in our paper.

P C3: Methods: This work is highly incomplete and far from a scientific study. You can easily tell with by the fact that there is no methods-chapter. There is an utmost need to include grain size analysis, microorganisms, sediment composition and other suitable methods.

The fact is that our method is not a sedimentary method. We found a thick layer of sandy sediments whose source could be any type of strong and energetic overwash event, like a hurricane or a tsunami. The excavation took place downtown and lasted only 3 weeks (in November, ie, during the rainy season). We began our investigation when the deepest layers were reached, a couple of days before the trenches was filled, and as mentioned previously it was not possible to keep them out of water all the

time. In this initial version of our paper, we argue that archaeological analysis, which indicates that the deposits took place between 1726 and 1770, and the determination of the origin of the sand in the mouth river are sufficient to attribute this deposit to 1755 Lisbon event. However, thanks to the observations of the reviewers, we understand that sedimentary data have to be published within this article, as they can be useful for the tsunami community.

Detailed investigations with all the suitable methods will be conducted when the next excavation will occur in Fort-de-France (in addition with sites 1, in 2012, and 2, in 2015, there has been only 2 archaeological excavations in Fort-de-France, in 1998 and 2003, during all the history of the town).

P C3 §5 "The material is siliciclastic and therefore you could check its age using OSL-dating."

We join to this reply an OSL (optically stimulated luminescence) dating report (Guerin, 2016, pers. comm. from M. Laforge) for a sample from Hotel de Police (site 2): no result was obtained (Figure 10).

P C3: It is ridiculous that you state (p. 6, lines 18-20) 'Our results indicate that the use of all available geological methodologies and collaborative studies with historians and archeologists might enable us to improve our historical tsunami catalogs in the future, thus helping the preparedness of tsunami hazard plans for coastal communities' as you have not applied any of the available geological methods!

We'll change the beginning of the sentence.

P C4: There is no information on the height above sea level of the excavation site and the event unit. How far is the site from the former coastline?

The height above sea-level is shown Fig. 2, d) in our paper and also indicated in the caption. As seen in Fig. 4 of the paper, the present-day distance to the shore is around 250-300m, and was of the same during the XVIII century. We'll add a mention of that

in the text.

P C4: Fig. 4 seems to show that the site is actually within one of the channels – is that true?

On 1726 map, there is indeed a channel arriving from a pound to the north-east, it then bypasses our parcel by the south, continues southwestward through the city (Figure 11). On 1766 map (Figure 12), there is no more channel. However, even when these maps exist, we cannot totally rely on them, 1) because everything is not drawn (for example, the pink parcels only refers to private owner and not to a building, whereas the constructions should be drawn when it is a public property, but it is not always the case) 2) sometimes there are not fully updated (for example, if it is a map done for specific purpose, like a representation of the fort), and 3) sometimes the maps describe a project not yet realized and it is not always specified. We think that the channels exist in 1726, as they do not look like a project. The time when they disappeared is not sure.

P C4: p. 6, line 6 and 15 'paleotsunami' – this is not the correct term. Paleotsunami units are lithified. Thus historical tsunami deposit would be the correct term.

Referee #3 is right, this is an historical event, we'll change it.

P C4: p. 4, lines 1-2 'and the sand particles sub-horizontal lamination (Fig. 3c) shows that the water enters the site and the sediments deposit slowly, and that there is no contribution from upper backflow.' This hardy makes sense. If the sediment particles slowly settle from the flow, normal (suspension) grading will be produced. Lamination is not a sedimentary structure that represents slow deposition! Instead lamination is caused by changes in the supply of sediment, most of all in terms of grain size and grain density.

We'll clarify this sentence. What we wanted to say was that the U-shape geometry of the building implies that there was not retreat wave(s) coming from the upper areas. However, we were no able to evidence grading: in the dark grey layer, the sand is very

homogeneous and the depth of the rounded tile pebbles does not indicate grading, and in the light grey layer is too thin to show grading.

In "Identification of tsunami deposits in the geologic record; developing criteria using recent tsunami deposits" by Peters and Jaffe, there is a paragraph on grading indicating that "Normal grading was common in tsunami deposits. ... Massive (ungraded) tsunami deposits were also common. Massive or normally graded sections may only be locally present."

P C4: p. 3, line 32 'The black layer globally follows the light grey layer, indicating a coeval deposition.' The deposition is not coeval, but successive otherwise one layer would not lie on top of the other.

We used coeval with the meaning of during the same event, and the fact that this was two successive deposits was implied. We'll change this formulation.

P C4: Attribution to the 1755 event: It is stated that the event unit was deposited in between 1726 and 1783 and thus I may resemble evidence of the 1755 tsunami. However, there was another tsunami in 1767 (see chapter 3). The 1767 tsunami was a regional event and thus could have had an even greater impact than the 1755 far field tsunami.

We have chosen between 1755 and 1767 events following the reasoning indicated in the paper: "In Martinique, numerous historical records –of 1755 event– report 1 to 3-m height waves in all the coastal areas, including Fort-de-France (Roger et al., 2011). Respectively, the Barbados earthquake of April 24th, 1767 generated a local tsunami (O'Loughlin and Lander, 2003; Lander et al., 2003) and 3-feet waves were observed only on the eastern coast of Martinique (see a newspaper extract in http://tsunamis.brgm.fr). It is doubtful that the impact of this latter event, which occurred only 12 years after the notable 1755 tsunami, would not have been reported in Fort-de-France." Being more precise, we should also mention the 1761 tsunami, but we ruled it out as it was not observed in Martinique. However, we'll add this 1761 event

in our discussion.

P C4: In addition, a storm event is excluded, even though large storms/hurricanes are frequently hitting the region. There was a severe hurricane in 1780 which killed thousands of people on Martinique and I do not see any argument why the described event unit could not be likewise related to this event.

We do not ignore the major hurricane of 1780, which devastated all the Lesser Antilles and a part of the Greater Antilles, from St Vincent to Jamaica. The analysis of historical report from the construction indicate that the Court of Appeal and the Police Station were built before 1770.

At site 1, the analysis of historical reports from the construction indicates that the Court of Appeal was built before 1774. We found this information in Marion (2000): in 1763, M. Daux, the owner of the building, is nominated tax collector by the king. The extension of the building is done around 1770. It is during this construction period that the overwash occurred. In 1774, following troubles with the law, M. Daux's goods are seized by the king and his residence in Fort-Royal is transformed in Courthouse with no indication of specific construction.

Results from site 2 analysis (Naveta et al., 2016) indicate that the embankment began after 1761 and not later than 70. In site 2, when reached, the deposit layer is everywhere above the mangrove and just below the first embankment.

We will scrupulously reproduce the historical information in the revised version of our article to clearly exclude 1780 hurricane.

P C5: Fig. 4: if the site is in or close to one of the historical channels, the settling needs to be interpreted differently. Backwash will use preexisting channels or topographic lows and you need to discuss the deposition of the unit in terms of this setting.

The historical center of Fort-de-France was built over a mangrove swamp and is flat. The mean altitude is 1-1.5m asl. There is no topographic lows nor highs. Because the

sandy layer is similar to the sand of the river mouth, we suspect a tsunami induced bore to carry the major part of the deposits. According to the urbanization indicated on the XVIII century maps and to the topography, we are not sure that the fact that these channels exists or not at the moment of the flooding would notably change the propagation of the incoming flow from the large Madam River (on the left of Fig 11). For the backflow, the most logical paths should be direct access to the sea or eventually by the channels.

We'll add a figure with the present-day topography, and will also include an historical map, in the geomorphic and climatic description of the area. We thank all the referees who all noted the lack of detailed description of the studied area.

P C5 Thickness and event magnitude (p- 4. Lines 16-19): The thickness of a tsunami deposit is directly related to the amount and grain size of available material, in addition to the topography (thicker units in topographic lows). There is no way to relate the magnitude of inundation or even of the event to sediment thickness!

We'll add a discussion on the parameters influencing the thickness of the deposit.

P C5: The authors state (p. 6, lines 14-15) 'This indicates that the tsunami deposit thickness used in tsunami modeling is a parameter that must be carefully checked in order to avoid overestimation of paleo-tsunamis and to correctly assess the tsunami hazard.' This is no new finding, but was already stated in several publications (e.g. Spiske et al., 2013). I do not see why the authors start the discussion of sediment thickness and inverse modelling because they did not do any inverse modelling. Roger et al. (2011) did forward modelling, but these results would only support the results of an inverse modelling approach.

This sentence appears in the conclusion, following the description of the type of sediment transport, mainly related to a tsunami bore. We might have been more explicit in the discussion and in the conclusion: in inverse modeling, if the 8-cm thickness found in Fort-de-France in used as input while the sediment transport in the river is not computed, the results might not be correct, whereas if 1cm is used, the results will be better. In forward model, if the river effect is not integrated, the observed 8-cm thickness will not be reproduced.

P C5: p. 6, lines 13-14: What exactly, in your opinion, is the difference of the tsunami wave front and a tsunami-induced bore? Explain what the differences in flow are and how they influence the mode of transport and deposition. What about the role of successive waves of the tsunami wave train?

We'll add some explanation regarding the capacity of a tsunami bore to transport higher volume of sediments, when compare to a direct wave. However, our goal was just to recall that a river in low lying area increases the risk: it is already known, but in risk assessment in Martinique and probably in other tropical islands, this is not taken into account, in particular as the Madam River is not considered as an important river, which is correct most of the time, but what should be wrong during the wet season (and 1755 event took place the 1rst of November, during the rainy season).

P C5: p. 6, lines 3-4 '1755 tsunami deposits can also be found in locations in eastern coastal American cities' – where should that be? Do you think that for example the US east coast was hit by the 1755 tsunami?

In 2014, exercise CaribeWave/Lantex was based on the Lisbon scenario, with a M8.5 earthquake. Although the parameters of the 1755 seismic source are still a matter of debate, with the source used during the exercise CaribeWave/Lantex 2014, the eastern coasts of Northern America and the north-eastern coasts of South America were impacted (Figure 13).

P C5-C6: Language: "The language is in many parts very poor..."

We apologize for that. A preliminary version was reviewed by an English colleague, but later, a lot of modifications were done. We'll ask to a colleague whose mother tongue is English to review our final version.

P C6: Fig. 1: runup height is given only as a number without unit. Add 'meters'

Will be done.

P C6: Fig. 2: add arrows that indicate the direction of the sea and the river.

Will be done.

P C6: Fig. 3: add arrows that indicate the direction of the sea and the river. Sediment photo needs to be much larger and resolution needs to be much higher. Lamination is described in the text and you need to prove this in the photo!

Will be done.

P C6: Fig. 4: 4a has no scale! 4a and 4b need to be much larger.

Scale appears on Fig 4b and is the same for 4a. We'll add a scale on both pictures and enlarge them.

P C6: Fig. 4: Again, just a visual comparison is worth nothing. Add more photos of the white layer and the proposed marine source material. Add lines for the topography. Otherwise the course of the inundation limit is hard to understand.

Will be done.

P C6: Tab. 1: What means 'H'? 'obs.' means 'observed'? Add explanation of the abbreviations to the captions.

Will be done.

References:

Exercise Caribe Wave/Lantex 14: A Caribbean and Northwestern Atlantic Tsunami Warning Exercise, Portugal Scenario, 26 March 2014, volume 1: Participant handbook [IOC/2013/TS/109VOL.1] Marion, G. G. (2000) : Administration des finances en Martinique : 1679-1790, L'Harmattan Ed. Navetat M., Nadeau A., J. Anctil, V. Bellavia, P. Butaud, A. Diederichs, F. Dieulafait, M. Laforge, F. Larre, C. Loiseau, S. Marchand,

L. Serra, S. Thomas and N. Tomadini, Fort-de-France (972), Nouvel Hôtel de Police, Rapport final d'opération archéologique (fouille préventive), Éveha – Études et valorisations archéologiques (Limoges, F), Hadès, 3 vol., SRA Martinique, 2016.

| | | | | |
|---|---|---|---|---|
| -38 | -4 | Brazil | H | Ruffman (2006) |
| -63.23 | 17.63 | Saba | H | Lander et al. (2002) |
| -61.80 | 17.02 | Antigua | H | Lander et al. (2002); O'Loughlin and Lander (2003) |
| -61.28 | 15.55 | Dominica | H | Lander et al. (2002) |
| -60.96 | 14.74 | Martinique | H | Roger et al. (2011) |
| -61.06 | 14.60 | Martinique | H | Roger et al. (2011) |
| -61.38 | 16.22 | Guadeloupe | H | Accary and Roger (2010) |
| -59.55 | 13.25 | Barbados | H | Baptista et al. (1998); Lander et al. (2002) |
| -63.05 | 18.05 | St Martin | H | Lander et al. (2002) |
| -74.5 | 20.01 | Cuba, Santiago | H | Rubio (1982); Lander et al. (2002) |
| -64.95 | 32.30 | Bermuda | H | Ruffman (2006) |
| -69.38 | 19.16 | Rep. Dominican, Samana Bay | H | Lander et al. (2002) |

**Fig. 1.** List of 1755 tsunami observation. H stands for historical.

| | | | | | | | | | | | | | | | | | | | | | | | | | | | | | | |
|---|---|---|---|---|---|---|---|---|---|---|---|---|---|---|---|---|---|---|---|---|---|---|---|---|---|---|---|---|---|---|
| • | | MARTINIQUE (FRENCH TERRITORY) | | EPINETTE RIVER | 14.73500 | -60.96000 | 5588 | | | | | | | | | | | | | | 1 | | | | | | | | | |
| • | | MARTINIQUE (FRENCH TERRITORY) | | FORT-DE-FRANCE | 14.60000 | -61.08300 | 5608 | | | | | | | | | | | | | | 1 | | | | | | | | | |
| • | | MARTINIQUE (FRENCH TERRITORY) | | LA TRINITE | 14.73700 | -60.96300 | 5588 | | | | | | | | 1.20 | | 66.00 | 1 | | | | | | | | | | | 1 |
| • | | MARTINIQUE (FRENCH TERRITORY) | | LAMENTIN RIVER | 14.60000 | -61.01000 | 5601 | | | | | | | | .90 | | | 1 | | | | | | | | | | | |
| • | | MARTINIQUE (FRENCH TERRITORY) | | LE FRANCOIS (SAINT-FRANCOIS) | 14.61900 | -60.89000 | 5590 | | | | | | | | | | | 1 | | | | | | | | | | | |
| • | | MARTINIQUE (FRENCH TERRITORY) | | LE GALION | 14.72000 | -60.93000 | 5587 | | | | | | | | | | | 1 | | | | | | | | | | | |
| • | | MARTINIQUE (FRENCH TERRITORY) | | MARTINIQUE | 14.74000 | -61.18000 | 5607 | | | | | | | | 1.80 | | 1500.00 | 1 | 1 | F | | | | | 1 | | | | |

**Fig. 2.** From NOAA database ( https://www.ngdc.noaa.gov/nndc/struts/form?t=101650&s=70&d=7).

Table 1

*Historical observed tsunami data in the Lesser Antilles for 1 November 1755 event*

| Place | Lon. (°W) | Lat. (°N) | Run-up (m) | Maximum inundation distance (MID) (m) | Withdrawal depth (m) | Withdrawal distance (MWD) (m) | Source |
|---|---|---|---|---|---|---|---|
| **Martinique** | | | | | | | |
| La Trinité | 60.96 | 14.74 | 0.6 (3), 0.9 (2), 1.2ᵃ (4, 5), 9.0 (1) | 66 (2) | 9.0 (1) | 6ᵇ (2), 66 (4, 5) | (1) Anonymous (1755), (2) Lettée (1755), (3) Daney (1846), (4) Brunet (1850), (5) Ballet (1896) |
| Le Galion | 60.92 | 14.73 | Minor effects | | | | Brunet (1850), Ballet (1896) |
| *Le Robert* | 60.94 | 14.68 | *Nothing observed* | | | | *Brunet (1850), Ballet (1896)* |
| *Sainte-Marie* | 60.99 | 14.78 | | | | | *Brunet (1850), Ballet (1896)* |
| Saint-François (actual Le François) | 60.89 | 14.62 | Just mentioned | | | | Anonymous (1755) |
| Lamentin's river | 61.01 | 14.60 | The sea rise up in the rivers | | | | Brunet (1850), Ballet (1896) |
| Fort Royal's river (actual Fort-de-France) | 61.06 | 14.60 | circa 0.9 m more than normal | | | | Brunet (1850), Ballet (1896) |
| Epinette's river | 60.96 | 14.735 | | | | | Daney (1846), Brunet (1850), Ballet (1896) |
| Martinique | 61.00 | 14.60 | Just mentioned | | | | Affleck (1756) |

**Fig. 3.** From Roger et al. (2011) : Historical observation of 1755 tsunami in the Lesser Antilles

![Fig. 4 photograph of excavation site]

**Fig. 4.** View of site 1, the Court of Appeal, at the end of the excavation

[Figure]

**Fig. 5.** The deposit layer is the dark layer just at water level

[Figure]

**Fig. 6.** The water pumps working at site 2, so that we could study the deposit.

[Figure]

7 cm

0.5-1 cm

**Fig. 7.** View of the deposit. The yellow ellipses round the light grey deposit.

[Figure]

**Fig. 8.** : Uninterpreted zoom of Figure 7 where the light grey shelly layer at the bottom of the dark grey layer is visible.

**Mean grain size distribution**

[Chart showing two curves: "Pointe Simon: River mouth" (blue) and "Court of Appeal" (red). X-axis labeled 0.1, 1, 10, 100, 1000, 10000. Y-axis labeled 0 to 8. A box labeled "Zone de traçage" appears on the plot.]

**Fig. 9.** Mean grain-size ($\mu$m) distributions of the samples

[Figure]

**Fig. 10.** OSL dating results from Guerin, 2016 (Navetat, Nadeau et al., 2016). Left: Right: OSL decay from site 2 sand.

[Figure]

**Fig. 11.** Extract from the 1726 map. . Site 1 is surrounded with an ellipse. A complete map can be found at url: (http://anom.archivesnationales.culture.gouv.fr/ulysse/osd?id=FR_ANOM_13DFC140A&q=carte&coverage

[Figure]

**Fig. 12.** 1766 map with no more channel. Site 1 is surrounded with an ellipse. (http://anom.archivesnationales.culture.gouv.fr/ulysse/osd?id=FR_ANOM_13DFC291A&q=carte&coverage=Martinique,%20%C3%8E

**Fig. 13.** ATFM (Alaska Tsunami Forecast Model) maximum amplitude map for the Atlantic basin (from Exercise Caribe Wave/Lantex 14 handbook).